# Analysis of Coupled Seepage and Temperature Fields of Fractured Porous Rock Mass under Brine–Liquid Nitrogen Freezing

**Shanshan Hou [1], Yugui Yang [1,2,*], Yong Chen [3], Dawei Lei [2] and Chengzheng Cai [1,2]**

[1] State Key Laboratory for Geomechanics and Deep Underground Engineering, China University of Mining and Technology, Xuzhou 221116, China; TB20220015B4@cumt.edu.cn (S.H.); caicz@cumt.edu.cn (C.C.)
[2] School of Mechanics and Civil Engineering, China University of Mining and Technology, Xuzhou 221116, China; xhchen@cumt.edu.cn
[3] State Key Laboratory of Coal Resource and Safe Mining, China University of Mining and Technology, Xuzhou 221116, China; chenyong@cumt.edu.cn
* Correspondence: ygyang2009@cumt.edu.cn

**Abstract:** The existence of fracture flow has an undesirable effect on the creation of the frozen wall. Brine and liquid nitrogen combined freezing technology can ensure the safety of freezing engineering, reduce the construction period and save cost. Considering the permeability of the rock matrix, fluid exchange and Darcy–Stokes coupling effect between the rock matrix and fracture, a thermo-hydraulic model of the fractured porous rock mass under water seepage is herein established. The interfacial seepage field characteristics of fractured rock mass under different fluid flow models and interface conditions are compared. The numerical simulations of the initial brine freezing and liquid nitrogen reinforcement freezing are carried out. The results show that the overall permeability of fractured rock mass computed by free flow considering the Darcy–Stokes effect is greater than that computed by the Cubic law. The limit seepage velocity of the intact rock mass in brine freezing is 2.5 m/d, and that of fractured rock mass decreases to 1 m/d. The fracture aperture and groundwater seepage velocity are directly proportional to the closure time of the frozen wall. Liquid nitrogen freezing can seal water quickly and shorten the closure time of the frozen wall when the seepage velocity of the fractured rock mass is greater than the limit seepage velocity, and the rapid cooling of the upstream region plays an important role in the formation of the frozen wall in fractured rock mass.

**Keywords:** coastal foundation pit engineering; artificial ground freezing; fractured rock mass; water seepage; liquid nitrogen freezing

## 1. Introduction

With the further development of urban underground space construction, a large number of underground infrastructure projects will face high permeability and water-rich soil layers or soft rock formations. Due to the complex geological conditions and water flow effects, the stability of coastal foundation pits is more prominent. Water sealing and underground structure stabilizing are two key problems to be solved in the construction process. Artificial ground freezing (AGF) technology is often utilized when shields are required to support unstable water-bearing strata [1–5]. Multiple rows of vertical freezing holes are arranged to form a closed frozen curtain around the tunnel excavation area to resist water and soil pressure and isolate groundwater [6,7]. In engineering practice, groundwater flow has been found to have adverse effects on the formation of the frozen curtain. For example, water flow causes an uneven thickness in the frozen wall, and the unfrozen zone cannot be sealed when the water flow velocity is high [8–12]. The occurrence of fractures with a large water flow rate undoubtedly aggravates the adverse effect on the formation of the frozen wall. For instance, the frozen area is disturbed during the shield

tunneling process, leading to the formation of fractures and hydraulic channels in weak places in the frozen wall [13,14]; and continuous water flow scours the frozen wall resulting in the collapse of the frozen wall and eventually floods the shield [15,16]. Therefore, it is of great practical significance to study the effect of fracture water flow on the development of frozen walls.

The change of heat and moisture in rock masses under low temperature and water flow is complicated because the temperature field and moisture migration are mutually coupled and influence each other [17–19]. The migration of water causes the change of thermal characteristics of rock and soil masses, which affects the distribution of the temperature field; and in turn, the change of temperature obviously affects fluid density and viscosity and results in a change of permeability in rock masses [20]. Numerous suggestions have been proposed to describe the hydrothermal coupling mechanics in the AGF process. For example, Wang et al. [8,21] conducted a comprehensive and systematic study on the formation mechanism of an artificial frozen wall in a permeable stratum under high water seepage velocity, based on a large-scale water–heat coupled physical model test system, which provides references for the layout of frozen holes in a high-velocity permeable stratum. Sudisman et al. [22] studied the change of temperature distribution surrounding freezing pipes under the action of seepage by using infrared images, and realized the visualization of heat distribution. Zhang et al. [23] proposed an indirect thermal-acoustic coupling method by using ultrasonic in situ detection to study the evolution characteristics of the temperature field and the rules governing change of acoustic characteristics during the freezing process of a water-rich sand layer. Song et al. [24] carried out experimental studies on the dual-pipe freezing temperature field of fractured rock mass under water seepage, and the results show that fracture water seepage significantly delays the time of frozen wall closure, and the thickness of the frozen wall is also reduced.

Numerical simulations can predict the evolution of the frozen zones accurately and reliably and provide reasonable guidance for the AGF construction process [3]. Several studies have been conducted to model the AGF process numerically. For example, Feng et al. [25] simulated the hydrothermal coupling and phase transformation problems in the freezing process of fractured rock mass, and analyzed the influence of fracture aperture and fracture inclination on the freezing temperature field and seepage field. Li et al. [26] proposed a heat–moisture coupling model to predict the dynamic formation process of the freezing curtain by combining heat transfer, Richard's equation and the Darcy equation of porous media. Vitel et al. [27,28] proposed a thermo-hydraulic model of ground around freeze-pipes and analyzed the influence of different vertical fracture locations and hydraulic conductivity coefficients on the temperature field. The results indicated that water seepage conditions have impact on the ground freezing process, whether the flow is due to the regional hydraulic gradient or to permeable fractures located in areas near the frozen zone. Huang et al. [29] established a heat–fluid coupling model of fractured rock mass under the condition of low temperature freezing, taking into account the process of water–ice phase transformation and water flow in fractures. Chen et al. [30] analyzed the evolution law of freezing temperature field under the seepage of single-fractured porous media rock mass by regarding the rock as an impermeable matrix; in this study, the water exchange between the rock and unfilled fracture was not considered.

When fluid in fractured porous media passes through the fractures, it will be coupled with the surrounding porous media. Because the water flow flux of fracture is much larger than that of rock matrix during model tests. In the flow flux measurement, the total flow flux is usually regarded as the flow flux of fracture, thus ignoring the seepage of rock matrix despite this being clearly inconsistent with the actual engineering. For this reason, theoretical analysis and numerical simulation research on the fluid flow and heat transfer in fractured porous media considering the interface coupling effects between rock matrix and fluid has become a hot issue. Chen et al. [31] used a dimensional reduction model to model fractures in a two-dimensional region, and a differential method to simulate single-phase Darcy flow in porous media with two-dimensional fractures. In order to

study the role of natural fractures in porous media, Zuo et al. [32] combined an embedded discrete fracture model (EDFM) with a streamline simulation method to simulate natural fractures, and calculated the fluid flow tracks in rock matrix and fractures. Starting from a mathematical flow model at the microscopic pore scale, Huang et al. [33] conducted scale upgrading research based on the volume averaging method, and established a new set of coupled Stokes–Darcy flow interface conditions. Alazmi et al. [34] summarized five primary categories of fluid flow interface conditions and four primary categories of heat transfer interface conditions between a porous medium and a fluid layer. The difference in model calculation results under different boundary conditions was systematically analyzed. It was shown that, in general, variations in interface conditions have a significantly more pronounced effect on the velocity field than on the temperature field.

At present, there is limited research on fluid flow and heat transfer in artificial freezing of fractured porous media that considers the interface coupling effects between rock matrix and fluid. In the numerical simulations of AGF for fractured rocks under water seepage, cubic law is usually used to describe the water flow in the fracture. Alternatively, the fracture is considered as a strong permeability zone with higher permeability than that of the rock matrix, and Darcy's law is used to simulate the fluid flow in the fractures. However, if the permeable rock contains fractures, the permeability of the rock matrix, fluid exchange and coupling between the rock matrix and the fracture will affect the overall permeability of the fractured porous medium [35]. The cubic law does not consider the permeability of the rock matrix around the fracture, and therefore does not adequately describe the permeability characteristics of fractured porous media. Moreover, in practical engineering experience, when water gushes along the cracks, the water flow velocity is relatively large, which does not satisfy Darcy's law [36]. In order to study the influence of high speed fracture-water flow on the frozen wall development in fractured rock mass, a thermo-hydraulic model of fractured rock mass is established in this current research. Based on the classical continuous boundary conditions and the slip velocity boundary conditions, the fluid exchange and hydrothermal coupling between rock matrix seepage and fracture water under low temperature are taken into account in the thermo-hydraulic model. Numerical simulations of double freezing pipes in fractured rock mass are carried out. The interfacial seepage field characteristics of fractured rock mass under different fluid flow models and interface conditions are compared. An analysis is undertaken of the influences of groundwater seepage velocity and fracture aperture on the interfacial seepage characteristics and temperature distribution of fractured rock mass under the initial brine freezing and liquid nitrogen reinforcement freezing.

## 2. Thermo-Hydraulic Coupling Model of Fractured Rock Mass

### 2.1. Governing Equations for Fluid Flow

#### 2.1.1. Rock Matrix

According to the theory of porous media, saturated porous rock subjected to freezing consists of three phases, namely solid particles (s), pour liquid water (l) and pour ice (i). During the freezing process, the hydraulic and thermal behavior of porous rock is described by the freezing characteristic curve, and governed by the conservation equations of mass, momentum and energy. Based on the volume averaging method, these equations can be obtained by adopting a Representative Elementary Volume (REV) in porous rock [3].

Based on the law of conservation of mass, in the absence of sources and sinks, the mass change between water and ice in the rock matrix is equal to the mass of unfrozen water flowing into or out of the rock matrix. The conservation equation of saturated porous medium under freezing conditions can be written as [29]:

$$n\frac{\partial(\rho_1 S_1 + \rho_i(1 - S_1))}{\partial t} + \nabla \cdot (\rho_1 \boldsymbol{u}_r) = 0 \tag{1}$$

where $t$ is the time; $n$ is the porosity of rock and considered as constant; $\rho_1$ and $\rho_i$ are the density of liquid water and ice, respectively; and $\boldsymbol{u}_r$ is the superficial water seepage velocity

in rock matrix, with $\boldsymbol{u}_{\mathrm{r}} = n\boldsymbol{u}_{\mathrm{ir}}$, and $\boldsymbol{u}_{\mathrm{ir}}$ represents the intrinsic average water seepage velocity in rock matrix. The term $S_1$ represents the saturation of liquid water, which is influenced by the freezing temperature and the pressure difference across the interface between liquid and ice, which is called the capillary pressure [6,27]. Van Genuchten proposed a parametric model for the isothermal, hysteretic unsaturated fluid phase content and hydraulic conductivity functions of unsaturated soils, which is also popularly used in representing the freezing characteristics of soil and rock medium [37]. Based on the Van Genuchten model, the freezing characteristic function of rock matrix can be expressed as:

$$S_l = \left[1 + \left(\frac{p_i - p_1}{P}\right)^{1/(1-m)}\right]^{-m}, \tag{2}$$

where $P$ and $m$ are parameters related to the pore structure. The terms $p_l$ and $p_i$ denote the pore water pressure and ice pressure, respectively. Assuming the chemical potentials of water and ice phases are in differential equilibrium, the Clausius–Clapeyron equation can be obtained to describe the water–ice phase change process as follows [38]:

$$\mathrm{d}p_i = \frac{\rho_i}{\rho_l}\mathrm{d}p_l - \frac{\rho_i L}{T_0}\mathrm{d}T, \tag{3}$$

where $L$ is the specific latent heat of water and ice phase change. Taking the atmospheric pressure and the temperature $T_0 = 273.15$ K as references, and integrating Equation (3) gives:

$$p_i - p_l = \frac{\rho_i L}{T_0}(T_0 - T). \tag{4}$$

Submitting Equation (4) into Equation (2), the relationship between the liquid saturation degree and temperature in freezing rocks can be expressed as [27]:

$$S_1 = \left[1 + \left(\frac{\rho_i L}{PT_0}(T_0 - T)\right)^{1/(1-m)}\right]^{-m}. \tag{5}$$

With $w = \frac{PT_0}{\rho_i L}$, Equation (5) can be written as:

$$S_1 = \left[1 + \left(\frac{T_0 - T}{w}\right)^{1/(1-m)}\right]^{-m}. \tag{6}$$

Darcy's equation is used to characterize the water seepage characteristics in rock matrix, and the pore water seepage velocity in rock matrix can be expressed as:

$$\boldsymbol{u}_r = -\frac{K_r K_r}{\mu_{\mathrm{lr}}}(\nabla \boldsymbol{p}_{\mathrm{r}} - \rho_1 \boldsymbol{g}) \tag{7}$$

where $\mu_{\mathrm{lr}}$ is the water viscosity in rock matrix; $\boldsymbol{p}_{\mathrm{r}}$ represents the water seepage pressure in rock matrix; $\boldsymbol{g}$ is the gravity acceleration vector; $K_{\mathrm{r}}$ is the intrinsic permeability of rock matrix; and $K_{\mathrm{r}}$ is the relative permeability of rock matrix, which describes the blocking effect of ice presence to the liquid water flow, and varies between 0 and 1. The relative permeability of rock matrix can be expressed as a function of water saturation as follows [39]:

$$K_{\mathrm{r}}(S_l) = S_l^{1/2}[1 - (1 - S_l^{1/m})^m]^2. \tag{8}$$

Considering the influence of freezing temperature, the viscosity of water in rock matrix can be expressed as [29]:

$$\mu_{\mathrm{lr}} = 2.1 \times 10^{-6} \exp\left(\frac{1808.5}{T}\right) \tag{9}$$

### 2.1.2. Fracture

For steady-state and incompressible laminar flow in fracture, the inertia force can be omitted in approximate treatment, that is, a Stokes-equation can be used to describe the flow characteristics in fracture:

$$\rho_1 \frac{\partial \boldsymbol{u}_f}{\partial t} = \nabla \cdot (\mu_{lf}(\nabla \boldsymbol{u}_f + \nabla \boldsymbol{u}_f^T)) - \nabla p_f \tag{10}$$

where $u_f$ is the water flow velocity in fracture and $\mu_{lf}$ is the viscosity of water in fracture. The water flow in fracture satisfies the law of conservation of mass, and can be written as follow [29,30]:

$$\frac{\partial(\rho_1 w_u + \rho_i(1 - w_u))}{\partial t} + \nabla \cdot (\rho_1 \boldsymbol{u}_f) = 0 \tag{11}$$

where $w_u$ is the content of unfrozen water in fracture. In order to obtain the stable and faster calculations, the following analytical function, a smooth approximation of the Heaviside function, is used to describe the unfrozen water in fracture [27]:

$$w_u = \frac{1}{1 + e^{-\varsigma(T-\theta)}} \tag{12}$$

where $\varsigma$ and $\theta$ are parameters related to the freezing characteristics of fracture water.

For the fracture water flow under low-temperature, the dynamic viscosity of water reflects the influence of the temperature field on the water flow velocity field. There is liquid water in the fracture when the temperature is above the freezing point. When the temperature drops below freezing point, ice crystals grow, and the fracture is filled with a mixture of ice and water. As the ice content increases, the viscosity of water in the fracture gradually increases, and the fracture water stops flowing after water is frozen. At this time, it is considered that the water viscosity in the fracture rises to infinity. Therefore, water viscosity in fracture can be expressed as a function of temperature as follows [40]:

$$\mu_{lf} = \begin{cases} \mu_{l0}, T > 273.15 \text{ K} \\ \mu_{l0}(1 + 2.5\,V + 10.25V^2 + 0.00273e^{16.6V}), 271.05 \text{ K} \leq T \leq 273.15 \text{ K} \\ \infty, T < 270.15 \text{ K} \end{cases} \tag{13}$$

where $\mu_{l0} = 0.001$ Pa·s is the constant liquid water viscosity at temperature 293.15 K, and V is the ice content, which can be written as:

$$V = 1 - w_u \tag{14}$$

### 2.1.3. Boundary Conditions at the Interface between Rock Matrix and Fracture

There is a small transition zone near the interface of porous media seepage and free flow in fractures. The transition zone can be considered to be a boundary layer zone where the fluid flow and heat transfer characteristics of a porous medium and a fluid adjust to one another [34]. The water flow coupling between fracture and rock matrix regions transfers physical quantities only at the interface. The physical model of the transition zone belongs to the Darcy–Stokes coupling problem, and the schematics of Darcy–Stokes problem are depicted in Figure 1. On the interface between fracture flow and porous media seepage, the fluid obeys the principle of mass flux continuity, continuity of normal stress, and the special Beavers–Joseph–Saffmann boundary condition on the tangential stress [33,41], which are:

$$\begin{cases} \boldsymbol{n}_f \cdot (\boldsymbol{u}_f - \boldsymbol{u}_r) = 0 & \text{on } \Gamma_{r-f} \\ p_f - \boldsymbol{n}_f^T \cdot \boldsymbol{\tau} \cdot \boldsymbol{n}_f = p_r & \text{on } \Gamma_{r-f} \\ -\boldsymbol{n}_f \cdot \boldsymbol{\tau} \cdot \boldsymbol{t} = \frac{\mu_{lr}\alpha}{\sqrt{t^T \cdot K_r \cdot t}}(\boldsymbol{u}_f - \boldsymbol{u}_r) \cdot \boldsymbol{t} & \text{on } \Gamma_{r-f} \end{cases} \tag{15}$$

where the subscripts f and r denote fracture water flow and rock matrix seepage flow, respectively; the terms $\boldsymbol{t}$ and $\boldsymbol{n}$ are the unit tangential and normal vectors to the interface

$\Gamma_{r-f}$; $\boldsymbol{\tau}$ is the stress tensor; and $\alpha$ is the velocity slip coefficient, which depends on the geometric and structural characteristics of the rock at the interface between rock-region and fracture-region.

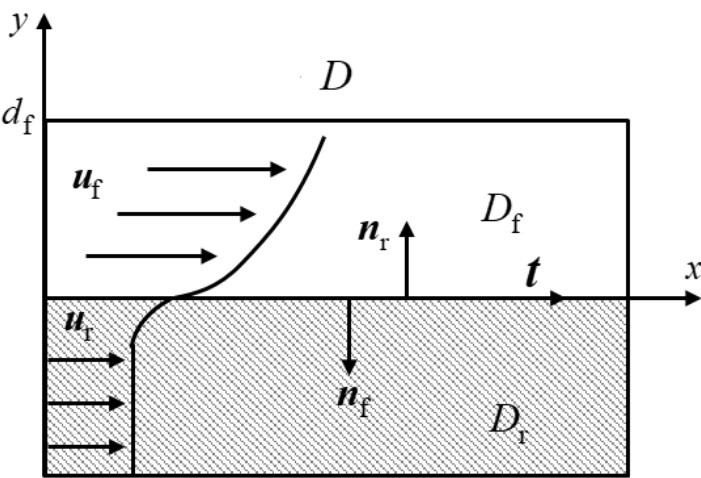

**Figure 1.** Schematic of Darcy–Stokes problem [33,41]. Subdivision of the domain $D$ into a free-flow subregion $D_f$ and a porous medium subdomain $D_r$ by an internal interface $\Gamma_{r-f}$.

*2.2. Governing Equations for Heat Transfer*

2.2.1. Porous Rock Matrix

Based on the local thermal equilibrium hypothesis, the conservation equation of energy for low temperature porous rock matrix, with consideration of the heat convection and water–ice phase transition, can be expressed as follows [29]:

$$(\rho C)_{eq}^{r}\frac{\partial T_r}{\partial t} + \rho_l C_l \boldsymbol{u}_r \cdot (\nabla T_r) + Q_r = -n_r \rho_l L \frac{\partial S_1}{\partial T_r} \tag{16}$$

where $T_r$ is the temperature of the rock matrix. The term $(\rho C)_{eq}^{r}$ is the equivalent heat capacity of rock, which can be written as [24]:

$$(\rho C)_{eq}^{r} = (1 - n_r)\rho_s C_s + n_r S_1 \rho_l C_l + n_r (1 - S_1)\rho_i C_i \tag{17}$$

where $C_s$, $C_l$ and $C_i$ are the heat capacity of rock particles, liquid water and ice, respectively. According to Fourier's law, the term of conductive heat flux $Q_r$ is proportional to the temperature gradient, and can be expressed as [3]:

$$Q_r = \nabla \cdot (-\lambda_r \nabla T_r) \tag{18}$$

where $\lambda_r$ is the equivalent heat transfer coefficient of rock, and can be calculated by using the geometric mean [3]:

$$\lambda_r = \lambda_s^{1-n}\lambda_l^{nS_1}\lambda_i^{n(1-S_1)} \tag{19}$$

with $\lambda_s$, $\lambda_l$ and $\lambda_i$ representing the heat transfer coefficient of solid particles, water and ice, respectively.

2.2.2. Fracture

Taking the REV of water flow in fracture as the research object, there are mainly three effects of convective heat transfer between water molecules, heat conduction of the fluid and thermal convection between the fracture water and the rock matrix that will cause the heat exchange of the REV. The convective heat transfer flow between water molecules can be expressed as [29]:

$$Q_{conv} = -\rho_l C_l \boldsymbol{u}_f \cdot (\nabla T_f) \tag{20}$$

where $T_f$ is the temperature of fracture. According to Fourier's law, the heat conduction of the fluid can be written as:

$$Q_{\text{cond}} = \nabla \cdot (-\lambda_f \nabla T_f) \tag{21}$$

where the term $\lambda_f$ is the equivalent heat transfer coefficient of fracture, and can be obtained by using the geometric mean:

$$\lambda_f = \lambda_1^{w_u} \lambda_i^{(1-w_u)} \tag{22}$$

The thermal convection between fracture water and rock is [29]:

$$Q_{\text{conv}}^f = h(T_f - T_r) \tag{23}$$

where $h$ is the convection heat transfer coefficient between rock and fracture water. For the fractures with aperture less than 1 cm, it can be considered that the surface temperature of the porous rock is equal to the temperature of the upper and lower surfaces of fracture water, and the heat transfers between fracture and rock is mainly related to the temperature gradient of the bedrock. From this, it can be obtained that [29]:

$$Q_{\text{conv}}^f = \lambda_f \frac{\partial T_r}{\partial n_{\text{fr}}} \tag{24}$$

Based on the assumption that the temperature of rock matrix surface is equal to that of fracture water, in the absence of source and sink terms for external heat exchange, the thermal balance equation of the REV in fracture flow can be obtained according to the law of conservation of energy:

$$(\rho C)_{\text{eq}}^f \frac{\partial T_f}{\partial t} + Q_{\text{cond}} + Q_{\text{conv}} = Q_{\text{conv}}^f - \rho_l L \frac{\partial w_u}{\partial T_f} \tag{25}$$

Submitting Equations (20), (21) and (24) into Equation (25), it can be seen that:

$$(\rho C)_{\text{eq}}^f \frac{\partial T_f}{\partial t} + \rho_l C_l \boldsymbol{u}_f \cdot (\nabla T_f) + \nabla \cdot (-\lambda_f \nabla T_f) = \lambda_f \frac{\partial T_r}{\partial n_{\text{fr}}} - \rho_l L \frac{\partial w_u}{\partial T_f} \tag{26}$$

where $(\rho C)_{\text{eq}}^f$ is the equivalent heat capacity of the fracture, and can be written as:

$$(\rho C)_{\text{eq}}^f = (1 - w_u)\rho_i C_i + w_u \rho_l C_l \tag{27}$$

## 3. Numerical Calculation Model of Freezing Fractured Rock Mass

### 3.1. Establishment of Model of Freezing Fractured Rock Mass under Water Seepage

There are many studies on the design parameters for underground infrastructure construction projects, and freezing effects on intact rock and soil mass. By contrast, there are relatively few such studies that research the freezing process of fractured rock mass under water seepage or consider the influence of fracture water on freezing effects. However, for rock mass with a large flow rate of fracture water leading to difficulties associated with grout plugging, artificial freezing technology is the only feasible construction method [29]. Therefore, it is very important to study the freezing effect of fractured rock mass under the coupling of water seepage and heat transfer. Numerical simulation methods have the advantage of reproducing engineering conditions that cannot be achieved by experimental methods. As depicted in Figure 2, in this study, the structural plane where the fracture is located is taken as the research object in the calculation model. It is assumed that there are two freezing pipes with a radius of 140 mm and a space of 1.4 m, and the fracture passes vertically through the middle of the two freezing pipes. The direction of water seepage is parallel to the vertical boundary from the top to the bottom. The hydraulic heads on the outlet boundaries are set to 0 m, the hydraulic difference between opposite surfaces is equal to the hydraulic head at inlet boundaries, and other boundaries are set as a no-flow boundary. The temperature of freezing

pipes is kept at 247.15 K during the freezing process. The physical and thermodynamic parameters of the rock involved in the study are listed in Table 1.

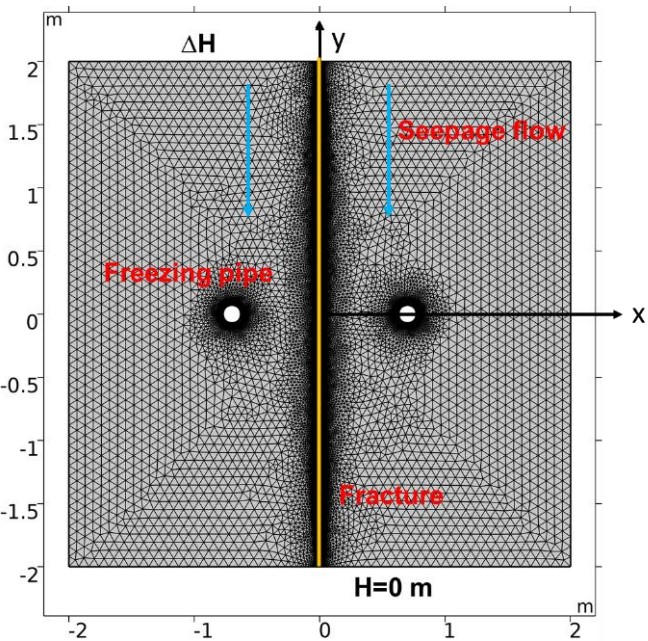

**Figure 2.** The calculation model for single−fractured porous media rock mass.

**Table 1.** Parameters related to the freezing process in fractured rock mass [27,29].

| Property | Symbol | Value | Units |
|---|---|---|---|
| Density of porous medium particles | $\rho_s$ | 2700 | kg/m$^3$ |
| Density of liquid water | $\rho_l$ | 1000 | kg/m$^3$ |
| Density of ice | $\rho_i$ | 917 | kg/m$^3$ |
| Thermal conductivity of porous medium particles | $\lambda_s$ | 4.3 | W/m/K |
| Thermal conductivity of liquid water | $\lambda_l$ | 0.6 | W/m/K |
| Thermal conductivity of ice | $\lambda_i$ | 2.2 | W/m/K |
| Specific heat capacity of porous medium particles | $C_s$ | 837 | J/kg/K |
| Specific heat capacity of liquid water | $C_l$ | 4200 | J/kg/K |
| Specific heat capacity of ice | $C_i$ | 2100 | J/kg/K |
| Latent heat of water and ice phase change | $L$ | 334 | kJ/kg |
| Porosity | $n$ | 0.41 | 1 |
| Intrinsic permeability | $K$ | $7.1 \times 10^{-15}$ | m$^2$ |
| Freezing point of water | $T_0$ | 273.15 | K |
| Parameter $\alpha$ | $\alpha$ | 0.6 | |
| Parameter $m$ | $m$ | 0.5 | |
| Parameter $\zeta$ | $\zeta$ | 4 | |
| Parameter $\theta$ | $\theta$ | 271.65 | |

*3.2. Model Mesh Independence Analysis*

The numerical model was simulated using the proposed governing equations. To solve such a highly nonlinear problem, the partial differential equations (PDEs) models of COMSOL were employed, and the corresponding partial differential equations are specified for the fracture region. The Newton method was used to solve the highly nonlinear system of governing equations, and the convergence criteria were set to 0.01. The computational model domain was meshed by free triangular mesh. Mesh independence tests were carried out for the present modeling to discover the optimum mesh size. In the modeling, the mesh

with the finest level was chosen to ensure accuracy of calculation. The mesh of the area of fracture and frozen pipes are refined locally. This leads to a total of 89,012 elements after meshing. In addition, the time step size is set to 0.1 d to provide accurate results, and the total time step is 300 days.

### 3.3. Validation of the Thermo-Hydraulic Model

The laboratory model conducted by Song et al. [24] was a numerical simulation based on the above governing balance equations to validate the proposed thermo-hydraulic model. The plane view of distribution of thermal couple along the fracture face and dimensions and boundary conditions of the model representing the experiment is depicted in Figure 3. There are four measuring lines located on the horizontal plane 20 cm away from the fracture plane to acquire the temperature distribution during the freezing process, in which S1, S2, S3 are parallel to the direction of water seepage, and F1 is perpendicular to the direction of water seepage. As listed in Table 2, the physical and thermodynamic parameters adopted in the simulation are the same as those involved in the study of reference [24]. Since the water seepage of rock matrix is ignored in the model test, the fluid flow velocity of fractured rock mass obtained in the test is the water flow velocity of fracture. Therefore, the same setting is adopted in the numerical simulation, and the seepage velocity of rock matrix is 0 m/d. The initial temperature of the rock sample is 23.7 °C, and the temperature of freezing pipes is stable at −25 °C. The diameter of the freezing pipes is 0.014 m.

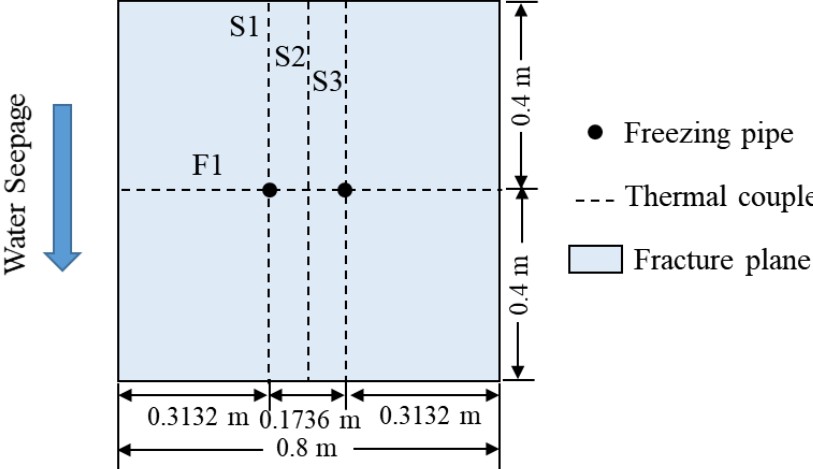

**Figure 3.** Distribution of thermal couple along the fracture face (adapted from reference [24]).

**Table 2.** Physical and thermodynamic parameters related to the rock sample (adapted from reference [24]).

| $n$ | $K$/(m/s) | $\rho_s$/(kg/m³) | $\lambda_s$/(W/m/K) | $C_s$/(J/kg/K) | $m$ | $T_0$ (K) |
|------|----------------------|------|------|------|-----|--------|
| 0.23 | $6.81 \times 10^{-8}$ | 2700 | 1.66 | 1430 | 0.5 | 273.15 |

In this study, a scenario of seepage flow with velocity 1.58 m/day has been simulated. The comparisons between the simulated and measured experimental temperatures along the parallel direction of water seepage (S1) are illustrated in Figure 4. It can be seen that, due to the heterogeneity of the experimental sample and the influence of the experiment environment, while the numerical simulation is in an ideal state, there are certain deviations between the simulated data and the measured temperatures, and the deviations are acceptable. The numerical simulation results adequately reflect the temperature decline trend in the fractured rock mass.

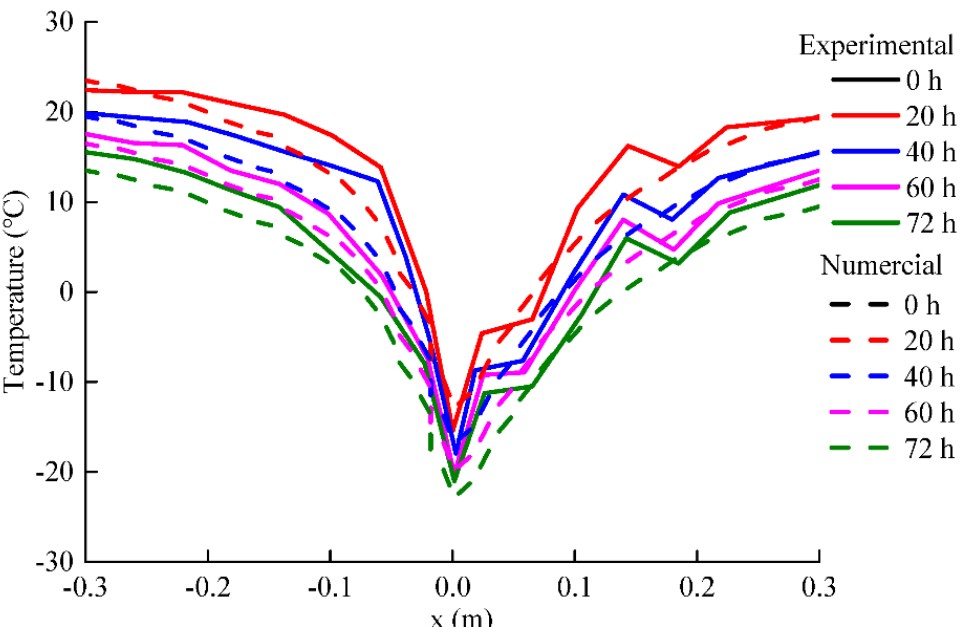

**Figure 4.** Comparisons between numerical solution and experimental measurements from reference [24].

## 4. Results and Analysis

### 4.1. Seepage Field Characteristics of Fractured Rock Mass

In order to study the influence of the fluid exchange at the interface between the rock and fracture, and the Darcy–Stokes coupling effects on the water seepage field of fractured rock mass, the interfacial seepage field characteristics of fractured rock mass under different fluid flow models and interface conditions were analyzed and compared.

#### 4.1.1. Fracture Apertures

The velocity characteristic of fractured rock mass with different fracture apertures are shown in Figure 5 for two cases: when the fracture flow is depicted by Cubic law (case 1), or as free flow while considering the Darcy–Stokes effect (case 2). It can be seen that the flow velocity and flux of fracture computed by free flow and considering the Darcy–Stokes effect are greater than that of fracture computed by Cubic law. As fracture opening increases, the differences between the two cases decrease (Figure 5a). This result is consistent with the research conclusion reached by Beavers and Joseph, namely: when the permeability of rock matrix is constant, the extra flow flux under slip velocity condition decreases with the increase in fracture opening [42] The water seepage velocity of rock matrix of case 2 is also larger than that of case 1 (Figure 5b). It can be seen that the overall permeability of fractured rock mass computed by free flow considering the Darcy–Stokes effect is greater than that of fracture computed by Cubic law. It is not possible to accurately depict the permeability characteristics of the freezing fractured rock mass, however, nor the temperature characteristics of the freezing fractured rock mass.

#### 4.1.2. Permeability of Rock Matrix

The velocity characteristic of fractured rock mass with different rock matrix permeability are shown in Figure 6, when the fracture flow is depicted either by Cubic law (case 1) or by free flow considering the Darcy–Stokes effect (case 2). It can be seen that rock matrix permeability has an insignificant influence on the fracture flow characteristics under a given water head. With the increase in rock matrix permeability, the flow velocities and fluxes of fracture in case 1 and case 2 barely change (Figure 6a). The seepage velocities and flux of rock matrix in case 2 are slightly larger than that in case 1. With the increase in rock matrix permeability, this gap gradually increases (Figure 6b). It is also observed that, even with the increase in rock permeability, the overall permeability of fractured rock mass

computed by free flow considering the Darcy–Stokes effect (case 2) are greater than that of fracture computed by Cubic law (case 1).

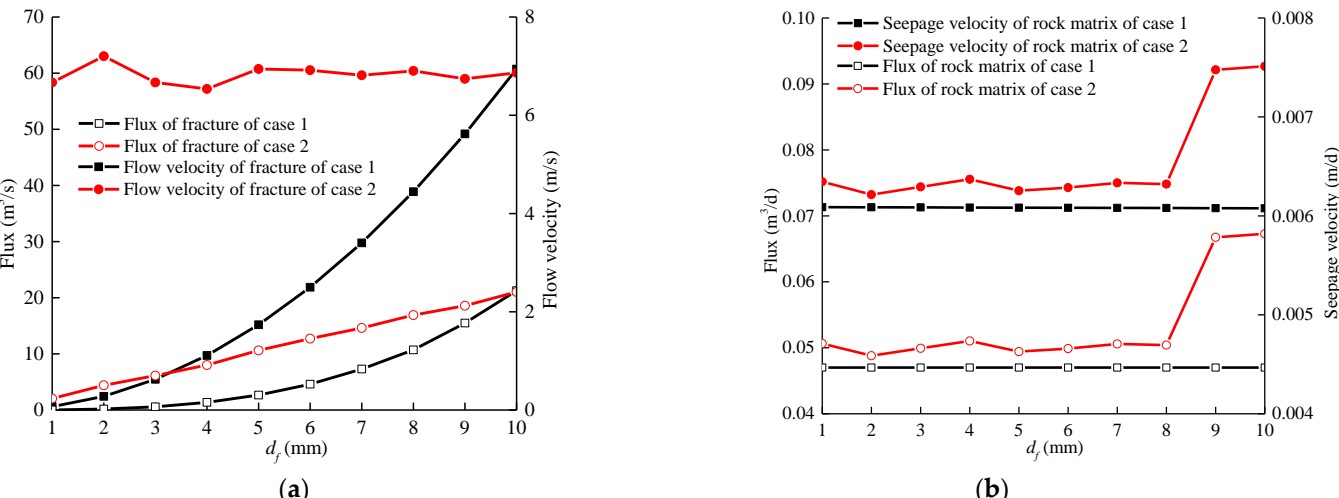

(a)

(b)

**Figure 5.** Velocity characteristic of fractured rock mass with different fracture aperture under $\Delta H$ = 3 m: (**a**) fracture and (**b**) rock matrix, where $d_f$ = fracture aperture.

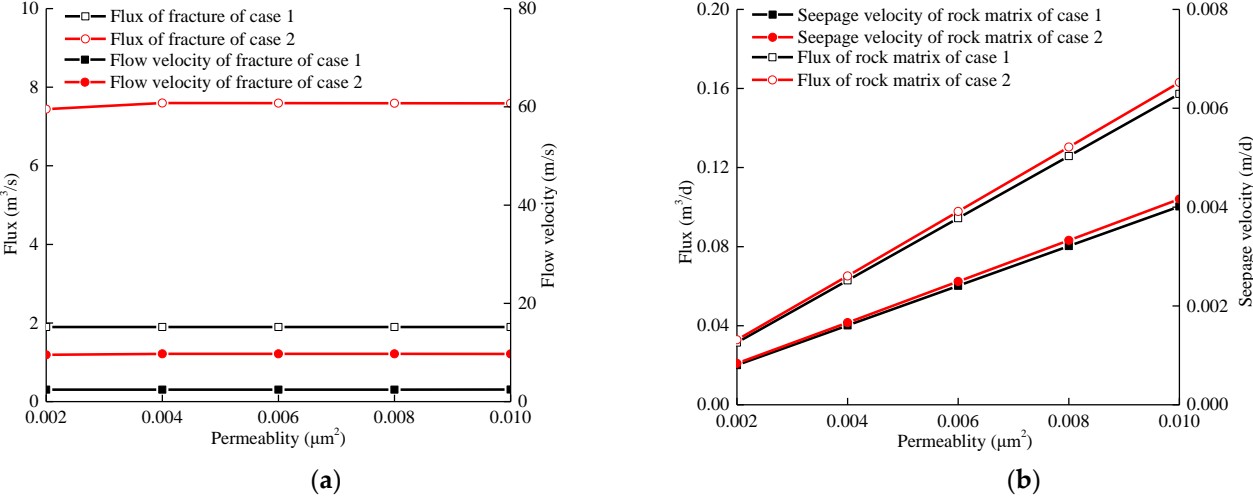

(a)

(b)

**Figure 6.** Velocity characteristic of fractured rock mass with different rock matrix permeability when $d_f$ = 5 mm ($\Delta H$ = 3 m and $\alpha$ = 0.3): (**a**) fracture and (**b**) rock matrix.

### 4.1.3. Velocity Slip Coefficient

The velocity characteristic of fractured rock mass with different velocity slip coefficients and various fracture apertures when the fracture flow is depicted by free flow considering the Darcy–Stokes effect are depicted in Figure 7. It can be found that the water flow velocity and flux through fractures increases with the rise in velocity slip coefficients. When velocity slip coefficient is greater than 0.3, the trend of increase slows down (Figure 7a). The same patterns emerge in the rock matrix (Figure 7b). It is also noticed that there is coupling effects between fracture aperture and slip coefficient on the fracture seepage velocity. When the slip coefficient is 0.01, the water flow velocity through fractures increases with size of fracture aperture. As the slip coefficient increases, the water flow velocity through fractures fluctuates with the increase in fracture aperture.

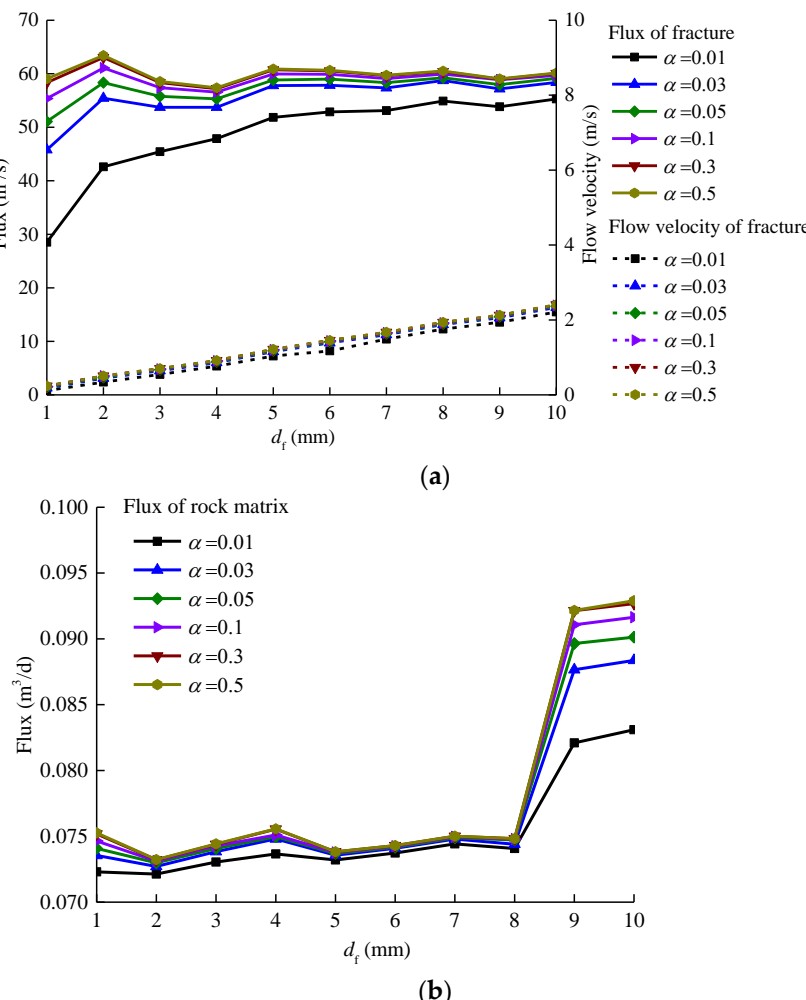

**Figure 7.** Flow velocity and flux through fractures in fractured rock mass with different apertures and different slip coefficients ($\Delta H = 3$ m, $d_f$ = fracture aperture): (**a**) fracture and (**b**) rock matrix.

### 4.2. Temperature Field Characteristics of Fractured Rock Mass

In order to study the development of the frozen wall in the brine freezing process of fractured rock mass, including consideration of the influence of the fluid exchange at the interface between the rock and fracture, the interfacial temperature, interfacial velocity and closure time were analyzed respectively.

#### 4.2.1. Interfacial Temperature

The interfacial temperature distributions of the intact rock during the freezing process under different seepage velocities are shown in Figure 8. The ordinate $y$ represents the interface coordinate, positive values representing locations upstream of the interface, and negative values representing those downstream of the interface. Comparing the interfacial temperature distribution of intact rock mass with different groundwater seepage velocity after freezing for 30 days, it can be seen that the temperature upstream of the interface is higher than that downstream, and that the cooling area downstream of the interface is larger. It can also be seen that the existence of groundwater seepage changes the original temperature field distribution of the rock mass. Due to convective heat transfer, water becomes the flow medium of heat, transferring the cooling capacity from upstream to downstream [21]. With the increase in water seepage velocity, the degree of asymmetry is greater (Figure 8).

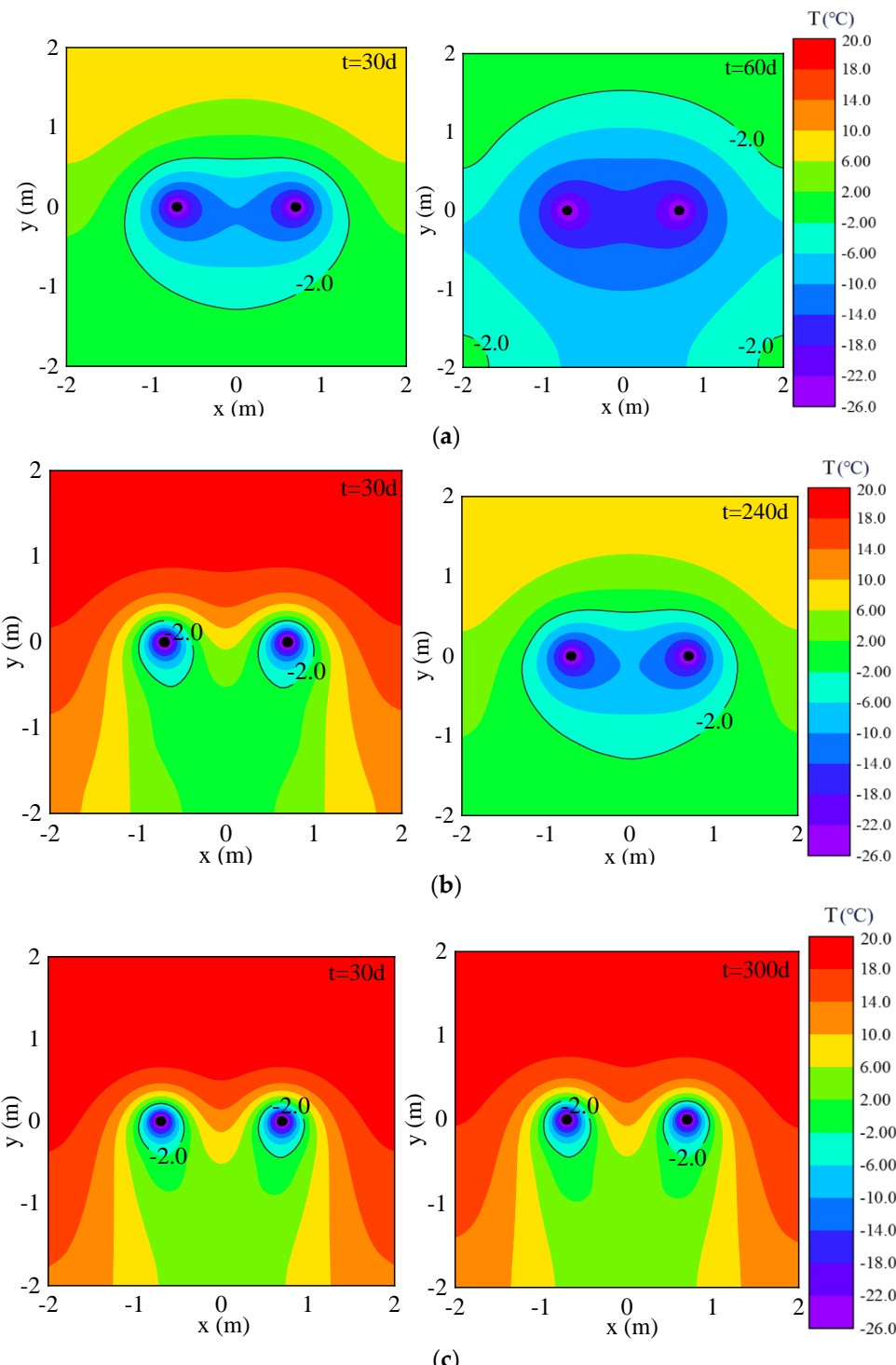

**Figure 8.** Interfacial temperature distribution developments of intact rock mass under different groundwater seepage velocities in the freezing process: (**a**) v = 1 m/d; (**b**) v = 1.5 m/d; (**c**) v = 2 m/d.

When the groundwater seepage velocity is 1 m/d, the frozen wall can intersect after 30 days of freezing (Figure 8a). When the groundwater seepage velocity is 1.5 m/d, the frozen wall intersects after 60 days of freezing (Figure 8b). When the groundwater seepage velocity is 2 m/d, the frozen wall can intersect after 240 days of freezing (Figure 8c). When the groundwater seepage velocity is 2.5 m/d, the frozen wall does not intersect after 300 days of freezing, and the thickness of the frozen wall barely increases even with the continuous increase in freezing time, making the intersection of frozen walls difficult. When

the groundwater seepage velocity is 3 m/d, the heat in the low temperature area is quickly taken away with the water flow, and the rock mass soon enters a heat balance state [4]. Therefore, the limit velocity of groundwater seepage in this study is 2.5 m/d.

From the temperature distribution during the freezing process at the interface of the fractured rock mass (Figure 9), it can be seen that as opposed to the "heart"-shaped distribution of temperature field of fractured rock mass that simulate the fluid flow in fracture by Darcy's law [25] the temperature distribution of fractured rock mass computed by free flow and considering the Darcy–Stokes effect presents a "butterfly" distribution before the intersection of the frozen wall. This is because the temperatures at fracture and rock around the entrance and exit of fractures decrease more slowly than elsewhere. In fractured rock mass, the flow velocity of fracture water is much higher than the seepage velocity of rock. Even if the rock matrix parts have been frozen, the fracture part still needs a long time to become frozen and for complete water plugging to happen. Moreover, due to the superposition effect of cooling capacity upstream and downstream, the cooling area downstream is larger than that upstream. When the fracture water is frozen and the frozen wall is crossed, the fractured rock mass rapidly cools down. The temperature distribution of the frozen body of fractured rock mass is symmetrical along the interface and presents as a "drum" distribution.

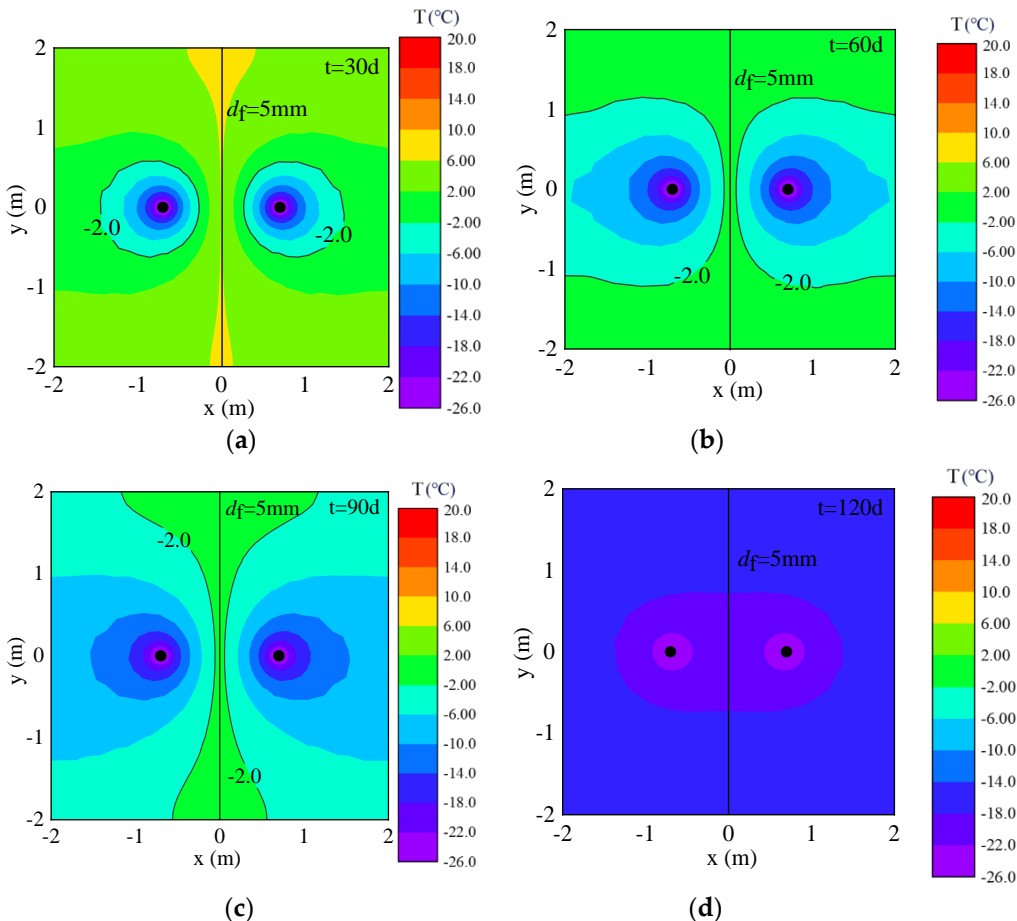

**Figure 9.** Interfacial temperature distributions of fractured rock mass during freezing with groundwater seepage velocity of 0.1 m/d: (**a**) t = 30 d; (**b**) t = 60 d; (**c**) t = 90 d; (**d**) t = 120 d.

### 4.2.2. Interfacial Velocity

This study considers that water plugging is complete when the velocity of water flow in the fracture is reduced to 0.05 m/d. Figure 10 depicts the seepage velocity distribution of rock at the interface ($y = 0$) during the freezing process of fractured rock mass with a fracture aperture of 5 mm under different groundwater seepage water velocities. The distances with

seepage velocity equal to zero represent the formation of frozen body thickness. Figure 11 shows the flow velocity distribution of fracture water (*x* = 0) during the freezing process of fractured rock mass with a fracture aperture of 5 mm under different groundwater seepage water velocities.

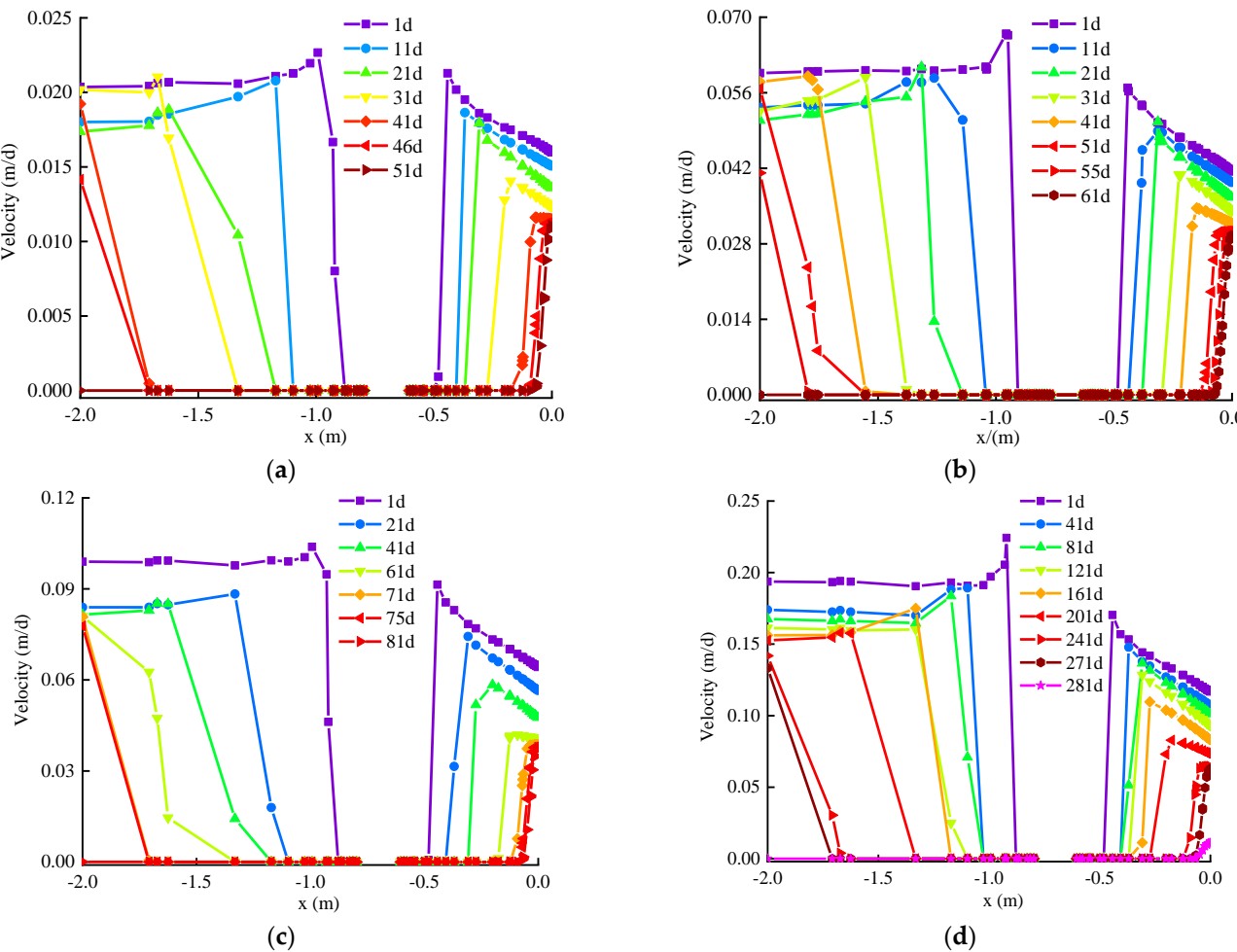

**Figure 10.** Variations in interfacial seepage velocity of rock in fractured rock mass with fracture aperture of5 mm during freezing, under different groundwater seepage velocities: (**a**) v = 0.1 m/d; (**b**) v = 0.3 m/d; (**c**) v = 0.5 m/d; (**d**) v = 1.0 m/d.

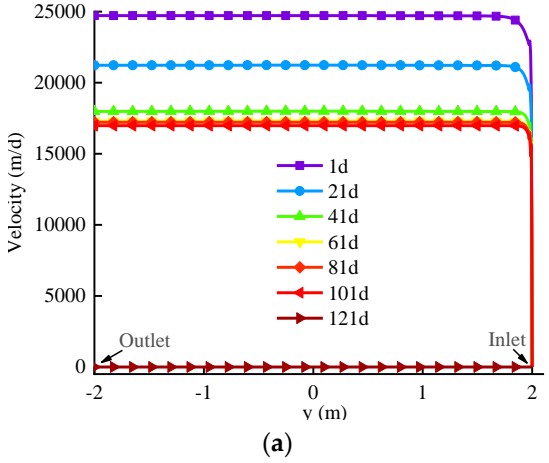

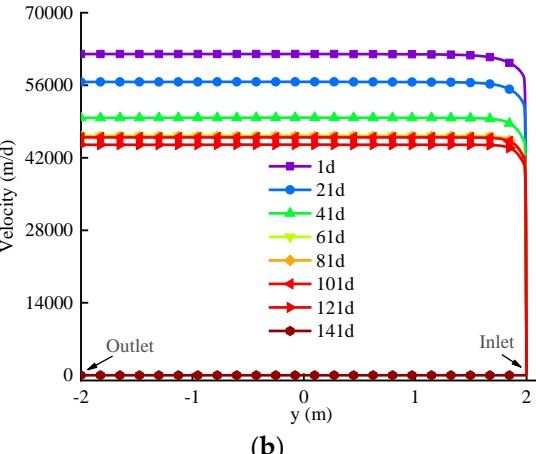

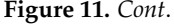

**Figure 11.** *Cont.*

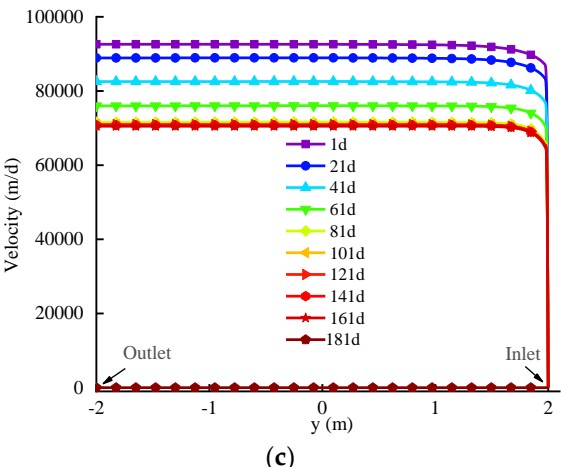
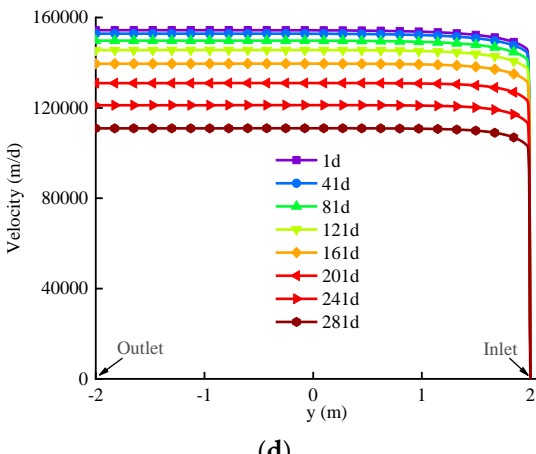

(**c**)
(**d**)

**Figure 11.** Variations in interfacial fracture flow velocity of fractured rock mass with fracture aperture of 5 mm during freezing under different groundwater seepage velocities: (**a**) v = 0.1 m/d; (**b**) v = 0.3 m/d; (**c**) v = 0.5 m/d; (**d**) v = 1 m/d.

The frozen area of rock increases rapidly, and the fracture water flow velocity decreases rapidly when the groundwater seepage velocity is 0.1 m/d. After 50 days of freezing, the seepage velocity of rock decreases completely to 0. Meanwhile the flow velocity of fracture water decreases to the target value after 70 days of subsequent freezing, and the frozen rock mass completes water plugging. When the groundwater seepage velocity is 0.3 m/d, the seepage velocity of rock drops to 0 after 60 days of freezing, and the flow velocity of fracture water drops to the target value after 80 days of subsequent freezing. When the seepage velocity of groundwater is 0.5 m/d, the seepage velocity of rock drops to 0 after 80 days of freezing. After 180 days of freezing, the flow velocity of fracture water decreases to the target value, and the frozen rock mass completes water plugging. When the groundwater seepage velocity is 1 m/d, the frozen area grows slowly. After 280 days of freezing, the seepage velocity of rock drops to 0, while the flow velocity of fracture water is still at a high level, which leads to the failure of intersection of the frozen wall. In conclusion, with the increase in groundwater seepage velocity, the freezing time of rock increases, and the presence of fractures increases the closure time of the freezing wall intersection because of the large water flow velocity in the fractures. The limit seepage velocity of the fractured rock mass is 1 m/d.

### 4.2.3. Development of Frozen Wall Thickness

The thickness of the frozen wall has an important effect on the strength and stability of the frozen wall. The frozen wall development rates of fractured rock mass with different fracture apertures under different seepage velocities are shown in Figure 12. The inside of freezing pipe represents the side near the fracture, and the outside of freezing pipe represents the side away from the fracture. It can be seen that with the increase in fracture aperture, the flow velocity in fracture increases and the water flux increases, so the heat carried away by fracture flow increases, which slows down the development of the frozen wall. In addition, the development rates of the frozen wall near the fracture are clearly slower than those far away from the fracture. When the water seepage velocity reaches the limit seepage velocity of 1 m/d and fracture aperture is greater than 7 mm, then the development of the frozen wall near the fracture stalls due to the large amount of cold energy brought away by the fracture water flow.

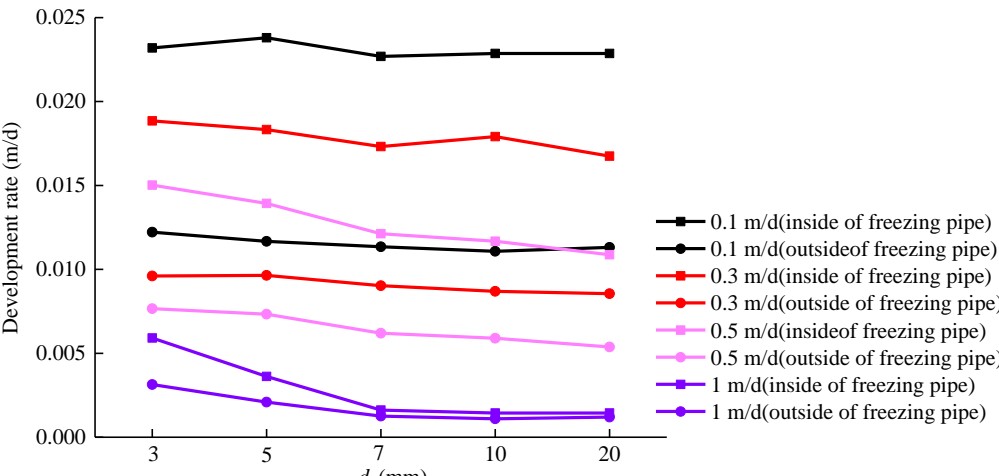

**Figure 12.** Development rate of frozen wall of fractured rock mass with different fracture apertures under different groundwater seepage velocities.

### 4.2.4. Closure Time of Frozen Wall

The freezing wall closure time distributions of fractured rock mass with different fracture apertures under different seepage velocities are depicted in Figure 13. It can be seen that, in the freezing process of fractured rock mass, fracture aperture and groundwater seepage velocity are positively proportional to the freezing wall intersection time. With the increase in groundwater seepage velocity, the freezing wall intersection time increases continuously. When the seepage velocity is 1 m/d and the fracture aperture is 3 mm, the freezing wall cannot cross closure within 200 d. However, adjusting the distance between freezing pipes has little effect on the temperature distribution of fractured rock mass. Moreover, the seepage velocity at the fracture is at an extremely high level, and material plugging measures cannot be taken at the fracture. It is then necessary to use liquid nitrogen freezing for plugging and reinforcing.

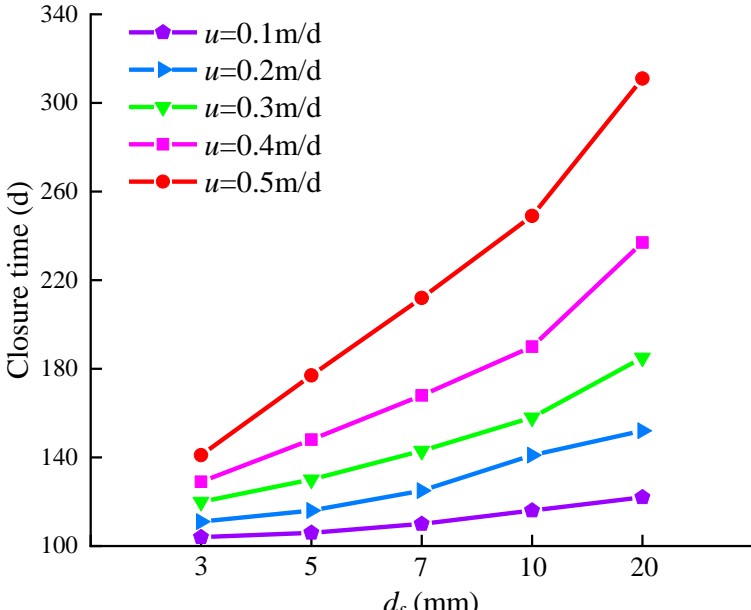

**Figure 13.** Closure time distributions of fractured rock mass with different fracture apertures under different groundwater seepage velocities, with $u$ = groundwater seepage velocity and $d_f$ = fracture aperture.

### 4.3. Liquid Nitrogen Reinforcement Freezing Process of Fractured Rock Mass

Liquid nitrogen freezing technology is widely used in engineering rescue because of its simple freezing equipment, fast freezing speed and high freezing wall strength [43,44]. When the flow velocity of underground water is large and conventional brine freezing cannot seal the unfrozen zone, the liquid nitrogen reinforcement freezing engineering method is often used in practice to seal the unfrozen zone, so as to form a closed frozen curtain. The liquid nitrogen freezing hole can be rearranged inside the original brine freezing hole [45]. In this study, the original freezing hole was chosen as the freezing hole for liquid nitrogen without restoring the original brine maintenance freezing, and the temperature of liquid nitrogen frozen wall was kept at 183.15 K during the liquid nitrogen reinforcement freezing process. The simulations of liquid nitrogen reinforcement freezing of fractured porous media rock mass were carried out after 200 days of brine freezing, and the interface temperature and fracture flow velocity were analyzed, as was the closure time of the freezing wall fractured rock mass with different apertures.

#### 4.3.1. Interfacial Temperature

The interfacial temperature distributions of fractured rock mass with different fracture apertures during the liquid nitrogen reinforcement freezing process are shown in Figure 14. It can be seen that the freezing wall closure time of fractured rock mass can be shortened quickly by liquid nitrogen reinforcement freezing. The interfacial freezing temperature distribution in fractured rock mass during the liquid nitrogen freezing process is similar to that of brine freezing. Before the freezing wall is enclosed, with the freezing temperature field showing a "butterfly-shaped" distribution, the initial cooling rate of the rock mass is rapid when the freezing wall is being formed, and the freezing temperature field shows a "drum" distribution after water plugging is complete. With the increase in fracture aperture, the water flux in fracture increases, and the heat carried away by fracture water increases. Therefore, the frozen wall of the fractured rock mass with larger fracture aperture takes more time to cross during the liquid nitrogen reinforcement freezing process. At the same time, it was found that due to the superposition effect of the upstream and downstream cooling capacity caused by the seepage of groundwater in the rock and the water flow through fractures, the cooling of the upstream region takes a long time. In other words, the rapid cooling of the upstream region plays an important role in the formation of the entire frozen wall in unfilled fractured rock mass. Therefore, when designing the freezing pipe layout, the installation density of the liquid nitrogen freezing pipe in the upstream area should be appropriately increased, so as to improve the freezing efficiency of the whole system.

#### 4.3.2. Development of Frozen Wall Thickness

The frozen wall development rates of fractured rock mass with different fracture apertures under liquid nitrogen reinforcement freezing are shown in Figure 15, with the water seepage velocity reaching the limit seepage velocity of 1 m/d. It can be seen that, compared with brine freezing, the development rate of the frozen wall is greatly improved under liquid nitrogen reinforcement freezing. The development rate of the frozen wall near the fracture is also significantly increased.

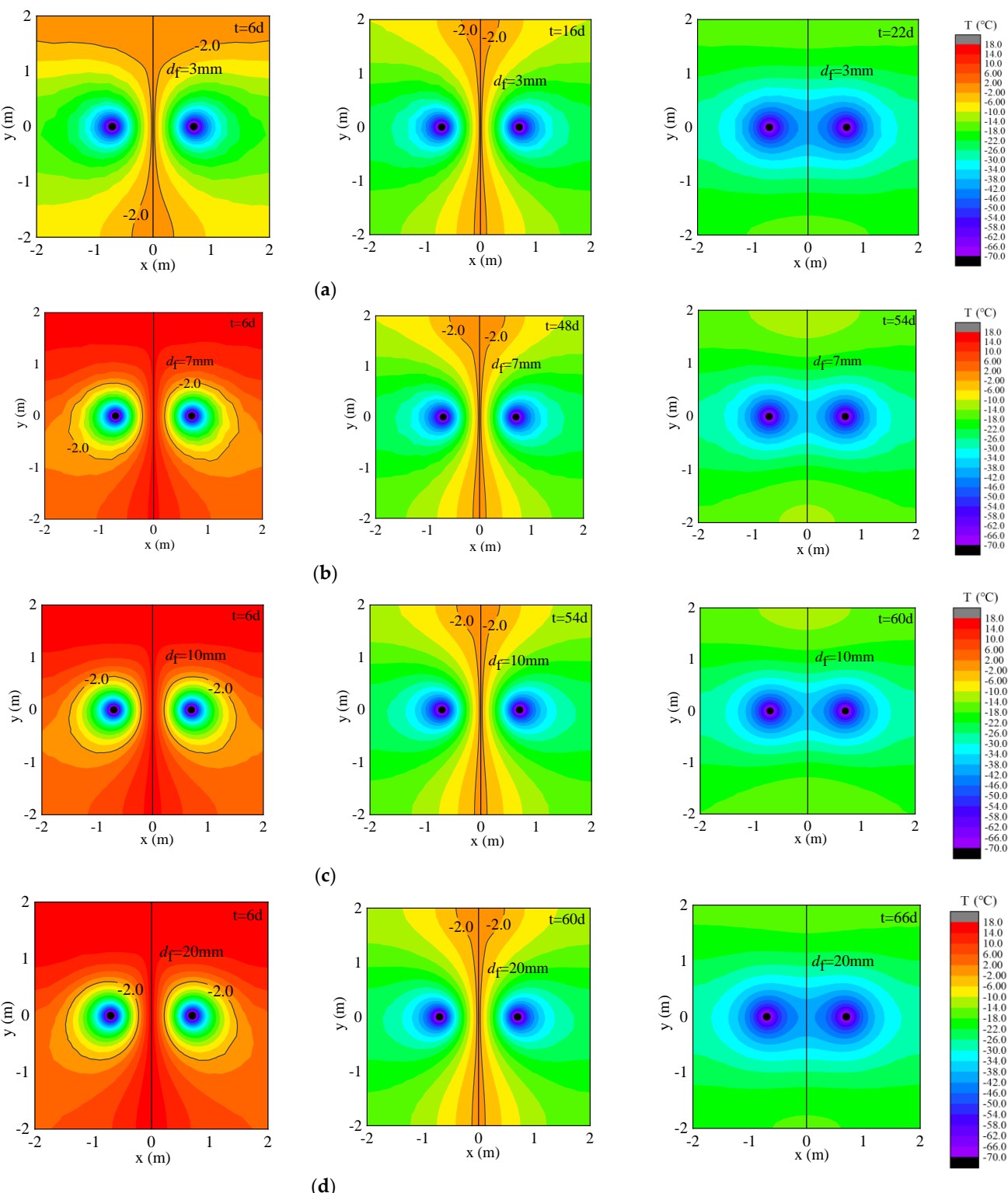

**Figure 14.** Interfacial temperature distributions of single-fractured porous media rock mass with different fracture apertures during the liquid nitrogen reinforcement freezing process: (**a**) $d_f$ = 3 mm; (**b**) $d_f$ = 7 mm; (**c**) $d_f$ = 10 mm; (**d**) $d_f$ = 20 mm.

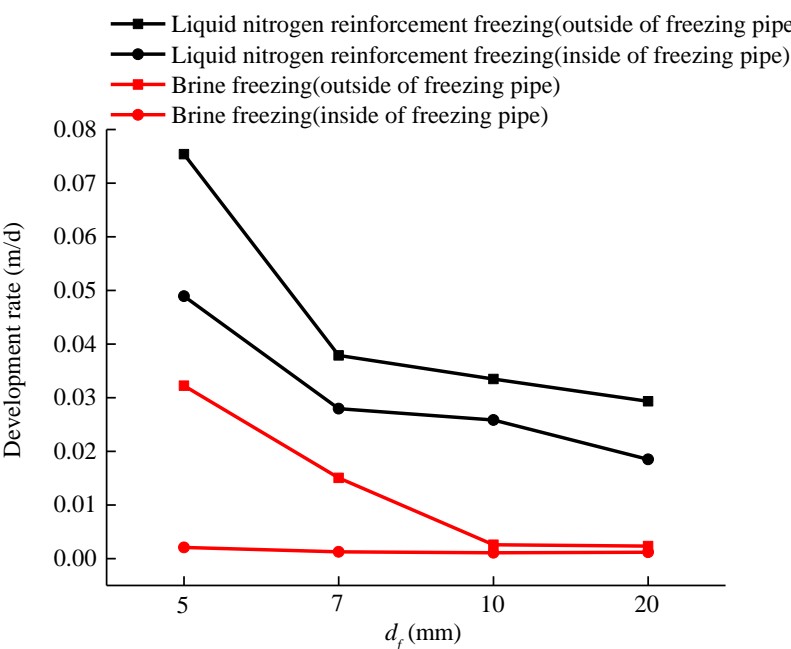

**Figure 15.** Development rate of frozen wall of fractured rock mass with different fracture apertures under liquid nitrogen reinforcement freezing.

### 4.3.3. Interfacial Velocity and Closure Time

The fracture flow velocities ($x = 0$) during the liquid nitrogen reinforcement freezing process of unfilled fractured rock mass with different fracture apertures are depicted in Figure 16. The relationship between the closure time of the frozen wall and the water flux in fracture are illustrated in Figure 17. It can be seen that the fracture water flow velocity decreases gradually with the cooling process and it rapidly drops to 0 when the temperature reaches the freezing point of fracture water, indicating that the fracture area is frozen. When the fracture aperture is less than 5 mm and the flux in unfilled fractures is less than 2000 m$^3$/d, the frozen wall can cross the closure within 30 days to complete water plugging. As fracture aperture increases, so does water flux in the fracture, and the time required for liquid nitrogen reinforcement freezing is longer. It is noted that the flux of fracture grows non-linearly as the fracture aperture increases when the fracture aperture increases from 5 mm to 10 mm. This phenomenon is caused by the Darcy–Stokes coupling effect between rock seepage and fracture water flow. From the relationship between flow velocity and flux in fractures of fractured rock mass with different apertures and different slip coefficients (Figure 7a), it can be seen that the flux in fracture does not increase linearly with the fracture aperture. Moreover, the coupling effect between seepage field and temperature field may have an influence on this non-linear relationship.

**Figure 16.** Interfacial velocity distributions at the fracture of single-fractured porous media rock mass during the liquid nitrogen reinforcement freezing process: (**a**) $d_f$ = 3 mm; (**b**) $d_f$ = 7 mm; (**c**) $d_f$ = 10 mm; (**d**) $d_f$ = 20 mm.

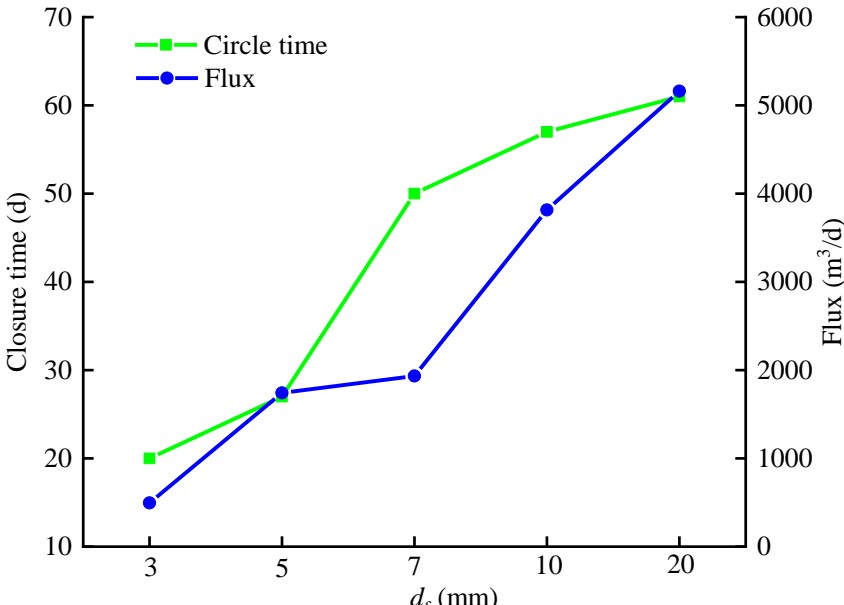

**Figure 17.** The relationship between the closure time of the frozen wall and the water flux in fracture.

## 5. Conclusions

The current experimental conditions are unable to accurately reflect the engineering environment of freezing fractured rock mass under water seepage. In the field of numerical simulation of freezing fractured rock mass under water seepage, there are only limited studies that consider the couple effect of Darcy flow of rock matrix and free flow through fracture. The current study establishes a universal hydrothermal coupling model for artificial freezing of fractured rock mass considering the permeability of the rock matrix, fluid exchange and the Darcy–Stokes coupling effect at the interface between rock and fracture. The interfacial seepage field characteristics of fractured rock mass under different fluid flow models and interface conditions were analyzed and compared. The effects of different groundwater seepage velocities on the velocity distribution as well as the development of frozen wall thickness and closure time of freezing wall intersection during the brine freezing process of fractured rock mass were analyzed. The freezing wall development in fractured rock mass was studied, notably when the brine freezing of the frozen wall was defective and later converted to liquid nitrogen reinforcement freezing. The following conclusions are drawn from the study:

(1) The overall permeability of fractured rock mass computed by free flow of fracture water considering the Darcy–Stokes effect is greater than that computed by the Cubic law. It is not possible to accurately depict either the permeability characteristics or the temperature characteristics of freezing fractured rock mass. Nevertheless, it is clear there are coupling effects between fracture aperture and slip coefficient on the seepage velocity of fracture.

(2) The numerical simulation results of temperature field distribution and development of fractured rock mass that fracture water flow depicted by free flow and considering Darcy–Stokes effect is different from that of fractured rock mass that considers the fracture as a strong permeability zone with higher permeability than that of the rock matrix, and used Darcy's law to simulate the fluid flow in fracture. Before the intersection of the frozen wall, the temperature distribution of fractured rock mass presents a "butterfly" distribution. After the frozen wall is closed, the temperature distribution of frozen rock mass is symmetrical along the interface and presents in a "drum"-shape distribution.

(3) Compared with brine freezing, the development rate of the frozen wall is greatly improved under liquid nitrogen reinforcement freezing, and the closure time of the frozen wall can be significantly shortened. The rapid cooling of the upstream region plays an important role in the formation of the entire frozen wall in fractured rock mass. Due to the coupling effect between fracture apertures and slip coefficients and seepage field and temperature field, the flux of fracture grows non-linearly as the fracture aperture increases when the fracture aperture increases from 5 mm to 10 mm.

In this study, the influence mechanism of water seepage on frozen wall formation of fractured rock mass was analyzed through the study of the interaction between water seepage of rock matrix and water flow of fracture and frozen wall formation. The proposed numerical simulation methods can better reproduce the conditions that cannot be achieved by experimental methods. The research results are helpful for proposing appropriate design methods and construction measures of freezing fractured rock mass, and provide the foundation for the expansion of the field of liquid nitrogen rapid freezing application.

**Author Contributions:** Methodology, investigation, formal analysis, writing—original draft preparation, S.H.; conceptualization, supervision, funding acquisition, Y.Y.; resources, investigation, Y.C.; software, validation, D.L.; resources, investigation, C.C. All authors have read and agreed to the published version of the manuscript.

**Funding:** This research was supported by the National Natural Science Foundation of China (41871063), the Fundamental Research Funds for the Central Universities (2019ZDPY18), and Independent Research Project of State Key Laboratory of Coal Resources and Safe Mining, CUMT (SKLCRSM19X015).

**Conflicts of Interest:** The authors declare no conflict of interest.

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
