# Peer review of "Analysis of Coupled Seepage and Temperature Fields of Fractured Porous Rock Mass under Brine–Liquid Nitrogen Freezing"

_jmse, doi:10.3390/jmse10060787_

Round 1

Reviewer 1 Report

The paper presents an investigation of seepage and temperature fields of fractured porous rock mass under brine-liquid nitrogen freezing. The document is organized and concise. English is good but grammatical errors, need to be addressed. The topic is relevant to the current research community. Please, see the comments and include modifications to the manuscript accordingly.

  1. The authors need to provide their institution email 163.com is not technically accepted.
  2. Please provide references for the data in all figures and tables, especially Table 1. The data is provided without any reference.
  3. Please read the paper a number of times and fix grammatical errors.
  4. Please provide a schematic diagram of fracturing with nitrogen freezing
  5. On page 7 authors mentioned that “freezing pipe is selected as research object”. How come pipe is used to validate a porous media rock
  6. Please provide the composition of rock used
  7. The authors must add a validation section and provide the validation of the model with experimental data, without model validation. The paper has no technical validity.
  8. The originality of the work is not obvious, please emphasize the originality that is not done before. Authors are using COMSOL, then please emphasize your contribution
  9. Please state a clear plan for your case study of rock fracturing and show the development process step by step, the authors need to present the development plan with all details and show the benefit the of applied/suggested technique
  10. Please show the benefits of the suggested report and compare them with the existing available techniques.

Author Response

Point 1: The authors need to provide their institution email 163.com is not technically accepted.

Response 1: Thanks to your valuable comments. Considering the Reviewer’s suggestion, we have changed the emails of all the authors to the institution emails (Please see Line#5-12).

Point 2:  Please provide references for the data in all figures and tables, especially Table 1. The data is provided without any reference.

Response 2: Thank you for your valuable comments. We are very sorry for our careless. According to the Reviewer’s comment, we have supplemented the references about the parameters related to the freezing process in fractured rock mass in the revised paper (Please see Line#299). For Figure 1., we have adjusted the annotation position of the literatures to make it better understood by readers (Please see Line#231).

Point 3: Please read the paper a number of times and fix grammatical errors.

Response 3: Thanks to your valuable comments. We have carefully checked and improved the English writing in the revised manuscript. And we have polished the manuscript with professional assistances in writing, conscientiously.

Point 4: Please provide a schematic diagram of fracturing with nitrogen freezing

Response 4: Thanks for your comments. We are sorry for the misunderstanding caused by our unclear statements. In this study, we investigate the temperature field of the brine freezing in fractured rock mass with large seepage velocity of groundwater. Then, the brine freezing is converted to liquid nitrogen reinforcement freezing on the basis of the original brine freezing pipes, which usually used in the practice engineering [Yuan, L. M. Application of liquid nitrogen to brine freezing reinforcement construction technology in subway tunnel. Low Carbon World 2020, 10, 167-168. 10.3969/j.issn.2095-2066.2020.05.104.]. Considering the Reviewer’s comment, we have rewritten the simulation scheme of liquid nitrogen reinforcement freezing process of fractured rock mass in the revised manuscript (Please see Line#495-503).

Point 5: On page 7 authors mentioned that “freezing pipe is selected as research object”. How come pipe is used to validate a porous media rock

Response 5: Thank you for your comments. We are sorry for the misunderstanding caused by our unclear statements. We have rewritten the sentence in the revised manuscript (Please see Line#288-289).

Point 6: Please provide the composition of rock used

Response 6: Thank you for your kind advice. In this study, a universal hydrothermal coupling model for artificial freezing of fractured rock mass is established and the distribution of freezing temperature field and velocity characteristics of fractured rock mass are studied based on the physical parameters commonly used in related studies [Vitel, M.; Rouabhi, A.; Tijani, M. Modeling heat and mass transfer during ground freezing subjected to high seepage velocities. Computers and Geotechnics 2016, 73, 1-15. https://doi.org/10.1016/j.compgeo. 2015.11. 014 and Huang, S. B.; Liu, Q. S.; Cheng, A. P.; Liu, Y. Z. A coupled hydro-thermal model of fractured rock mass under low temperature and its numerical analysis. Rock and Soil Mechanics 2018, 39, 735-744. https:// doi.org/10.16285/j.rsm.2017.0721]. We did not study the freezing effect of specific rock strata. In the future, we will carry out relevant studies combined with the specific engineering background. Thanks for your comments and suggestions once again.

Point 7: The authors must add a validation section and provide the validation of the model with experimental data, without model validation. The paper has no technical validity.

Response 7: Thank you for your instructive suggestions. To the best of our knowledge, limited scholars have carried out similar model tests of freezing temperature field of fractured rock mass under water seepage. However, the water flow flux of fracture is much larger than that of rock matrix in the model tests. In the flow flux measurement, the total flow flux is usually regarded as the flow flux of fracture, and ignoring the seepage of rock matrix, which is inconsistent with the actual engineering. The numerical calculation model in this study applying the seepage conditions of fractured rock in the actual engineering that considering the permeability of the rock matrix, fluid exchange and Darcy-Stokes coupling effect between the rock matrix and fracture. The current experimental conditions are difficult to truly restore the engineering environment, so it is unreasonable to compare the results with the existing model test results in the case of known deviations. In addition, this is also the advantage of numerical simulation, which can be better reproduced the conditions that cannot be achieved by experimental methods. In the future, we will carry out relevant experimental work to overcome the shortcomings of the existing model tests. At last, thank you very much for your comments and suggestions once again.

Point 8: The originality of the work is not obvious, please emphasize the originality that is not done before. Authors are using COMSOL, then please emphasize your contribution

Response 8: Thanks to your valuable comments. According to your helpful advice, we have strengthened the introduction to highlight our new idea (Please see Line#91-97 and Line#113-115). Besides, we have highlighted the contribution of the numerical simulation method employed in this study in the conclusion in the revised manuscript (Please see Line#582-584)

Point 9: Please state a clear plan for your case study of rock fracturing and show the development process step by step, the authors need to present the development plan with all details and show the benefit the of applied/suggested technique

Response 9: Thanks for your comments. We are sorry for the misunderstanding caused by our unclear statements. In this study, we establish a hydrothermal coupling model for artificial freezing of fractured rock mass under water seepage. The distribution of brine and liquid nitrogen reinforcement freezing temperature field and velocity characteristics of fractured rock mass are studied by numerical simulation. Considering the Reviewer’s comment, we have the supplemented the content about the simulation scheme of liquid nitrogen reinforcement freezing process of fractured rock mass in the revised manuscript (Please see Line#495-503). Besides, we have highlighted the advantages of the proposed model in the conclusion (Please see Line#567-573).

Point 10: Please show the benefits of the suggested report and compare them with the existing available techniques.

Response 10: Thank you for your instructive suggestions. According to your helpful advice, we have strengthened the advantages of the numerical simulation method used in this study compared with the similar model tests to study the temperature field and seepage field of frozen seepage fractured rock mass (Please see Line#330-332).  

Reviewer 2 Report

In general, the manuscript produces a very favorable impression. It contains a good analysis of the literature. The authors formulate a very interesting mathematical model of soil freezing, which undoubtedly has a novelty. However, there are a number of small remarks to the text:

Line 228 and line 235 - it is not clear why the temperature is in bold. Apparently this is a typo.

Figure. 3. It is not entirely clear why some of the red curves are oscillatory (especially for small d_f). Is it a numerical error or some kind of physics?

Figure 16. It is not clear why the flux grows so non-linearly as the fracture aperture increases. Especially in the range of 5-10 mm. Can the authors comment on this?

The Section 4 "Results and Analysis" lacks a deeper analysis of the patterns of freezing of cracked soils. The authors simply give a lot of figures and briefly describe what they depict. But there is no generalization of the presented data.

Author Response

Point 1: Line 228 and line 235 - it is not clear why the temperature is in bold. Apparently this is a typo.

Response 1: Thank you for your careful work. We are very sorry for our incorrect formula writing, and the incorrect formula writing has been revised (Please see Line#239, Line#247 and, Line#263 and Line#282).

Point 2: Figure. 3. It is not entirely clear why some of the red curves are oscillatory (especially for small d_f). Is it a numerical error or some kind of physics?

Response 2: Thanks to your valuable comments. This phenomenon is caused by the coupling effect of fracture aperture and slip coefficient on the flow rate of fracture water. Considering the Reviewer’s comments, we have rewritten the parts of 4.1.3. Velocity slip coefficient in the revised manuscript (Please see Line#346-355 and Figure 5).

Point 3: Figure 16. It is not clear why the flux grows so non-linearly as the fracture aperture increases. Especially in the range of 5-10 mm. Can the authors comment on this?

Response 3: Thank you for your comments. According to the Reviewer’s comments, we have supplemented the explanation of this phenomenon in the new manuscript  (Please see Line#549-557).

Point 4: The Section 4 "Results and Analysis" lacks a deeper analysis of the patterns of freezing of cracked soils. The authors simply give a lot of figures and briefly describe what they depict. But there is no generalization of the presented data.

Response 4: Thank you for your instructive suggestions. Considering the Reviewer’s comments, we have deleted Figure 8, Figure 9, and Figure 10 because of the repetitive content of its expression in the original text. Besides, we have added the content of analysis of the influence about the water seepage velocity and fracture aperture on the development of frozen wall thickness (Please see Line#460-472 and Figure 10) and the frozen wall development of fractured rock mass under liquid nitrogen reinforcement freezing (Please see Line#532-538 and Figure 13) in the revised manuscripts.

Round 2

Reviewer 1 Report

   The authors forget to address the following comments. Please address clearly

   The authors must add a validation section and provide the validation of the  model with experimental data, without model validation. The paper has no technical validity.

   The originality of the work is not obvious, please emphasize the originality that is not done before. Authors are using COMSOL, then please emphasize your contribution.

    Please state a clear plan for your case study of rock fracturing and show the development process step by step, the authors need to present the development plan with all details and show the benefit the of applied/suggested technique

Author Response

Point 1: The authors must add a validation section and provide the validation of the model with experimental data, without model validation. The paper has no technical validity.

Response 1: Thank you for your instructive suggestions. Considering the Reviewer’s comment, we have added the contents about validation of the proposed thermo-hydraulic model with experimental data in the revised manuscript (Please see Line# 328-356).

Point 2: The originality of the work is not obvious, please emphasize the originality that is not done before. Authors are using COMSOL, then please emphasize your contribution

Response 2: Thanks to your valuable comments. According to your helpful advice, we have strengthened the conclusion to highlight our new idea and the contribution of this study (Please see Line# 620-626 and Line#637-665).

Point 3: Please state a clear plan for your case study of rock fracturing and show the development process step by step, the authors need to present the development plan with all details and show the benefit the of applied/suggested technique

Response 3: Thanks for your comments. Considering the Reviewer’s comment, we have supplemented the content about the establishment step of model of freezing fractured rock mass under water seepage and highlighted the advantages of the applied method (Please see Line#289-304).